# Comparative Study of S2-Alar-Iliac Screw Trajectories between Males and Females Using Three-Dimensional Computed Tomography Analysis: The True Lateral Angulation of the S2-Alar-Iliac Screw in the Axial Plane

**DOI:** 10.3390/jcm11092511

**Published:** 2022-04-29

**Authors:** Haruki Funao, Kento Yamanouchi, Naruhito Fujita, Yukihiro Kado, Shuzo Kato, Nao Otomo, Norihiro Isogai, Yutaka Sasao, Shigeto Ebata, Yuko Kitagawa, Kota Watanabe, Hideaki Obara, Ken Ishii

**Affiliations:** 1Department of Orthopaedic Surgery, School of Medicine, International University of Health and Welfare, Narita 286-0048, Japan; yamaken0331@gmail.com (K.Y.); naruhito88@hotmail.com (N.F.); shuzokato800@gmail.com (S.K.); naoootomo@yahoo.co.jp (N.O.); n.isogai0813@gmail.com (N.I.); sasaospine@iuhw.ac.jp (Y.S.); ebatas310@gmail.com (S.E.); 2Department of Orthopaedic Surgery, International University of Health and Welfare Narita Hospital, Narita 286-8520, Japan; 17a1028@g.iuhw.ac.jp; 3Department of Orthopaedic Surgery, International University of Health and Welfare Mita Hospital, Tokyo 108-8329, Japan; 4Department of Surgery, Keio University School of Medicine, Tokyo 160-8582, Japan; kitagawa@a3.keio.jp (Y.K.); obara.z3@keio.jp (H.O.); 5Department of Orthopaedic Surgery, Keio University School of Medicine, Tokyo 160-8582, Japan; watakota@gmail.com

**Keywords:** sacropelvic fixation, S2-alar-iliac screw, percutaneous S2-alar-iliac screw, minimally invasive spine stabilization, minimally invasive spinal treatment

## Abstract

The S2 alar-iliac screw (S2AIS) is commonly used for long spinal fusion as a rigid distal foundation in spinal deformity surgeries, and it is also used in percutaneous sacropelvic fixation for providing an in-line connection to the proximal spinal constructs without using offset connectors. Although the pelvic shape is different between males and females, reports on S2AIS trajectories according to gender have been scarce in the literature. In this paper, S2AIS trajectories are compared between males and females using pelvic three-dimensional computed tomography (3D-CT) in a normal Japanese population. After resetting the caudal angulation in CT-imaging plane manipulation, the angulation of S2AIS was more lateral in the axial plane and more horizontal in the coronal plane in females. Mean distances from the midline to starting points of S2AIS tended to be shorter in females, whereas mean distances from the midline to the posterior superior iliac spine was significantly longer in females. We also found that there were positive correlations between the patients’ height and the maximal lengths of S2AISs, and the patients’ height and minimal areas of S2AIS pathways. Our results are useful not only for conventional open spinal surgery, but also for minimally invasive spine surgery.

## 1. Introduction

Sacropelvic fixation is widely used in various spinal conditions, including spinal deformities, tumors, and infections. The fixation provides a rigid distal foundation in the lumbosacral spine. Sacropelvic fixation is required to resist the flexion moments present at the lumbosacral junction to prevent fixation failures [1]. To date, various sacropelvic fixation techniques have been reported, such as Galveston iliac rods, Jackson intrasacral rods, Kostuik transiliac bar, iliac screws, iliosacral screws, sacroiliac buttress screws, and S2-alar-iliac screws (S2AIS) [2,3,4,5,6,7,8,9,10]. Currently, iliac screws and S2AIS are mostly used with relatively high fusion rates [4,7,8,9]. S2AIS is commonly used as a rigid and secure distal foundation in maintaining deformity correction and enhancing bony union at the lumbosacral junction, particularly in long spinal fusion to the sacrum. The S2AIS technique has several benefits, such as less soft tissue dissection, low-profile implants, and in-line connection to the proximal spinal constructs [7,8,9]. S2AIS is also applied to minimally invasive spine surgery, because it can be inserted percutaneously in line with the proximal spinal instrumentation without using offset connectors [8,9]. S2AIS is usually placed with a freehand technique or image guidance using either intraoperative fluoroscopy or computed tomography (CT)-based navigation. However, serious complications, such as neural, vascular, or bowel injuries, can occur due to a misplacement of S2AIS. It would be difficult for surgeons to insert percutaneous S2AIS without exposing anatomical landmarks; therefore, it is important to assess optimal trajectories of S2AIS in order to develop a percutaneous technique in minimally invasive spine surgery as well as open surgery.

Since pelvic size and shape are different between males and females [11], optimal S2AIS trajectories should be evaluated regarding gender differences. To date, there have only been a few reports that evaluate the trajectories of S2AIS according to gender [12,13,14,15]. Although optimal S2AIS trajectories are determined in the transverse plane using CT-imaging plane manipulation adjusting caudal angulation, the lateral angulation of S2AIS should be measured in the axial plane after resetting the caudal angulation. However, the lateral angulation of S2AIS has been measured only in the transverse plane without resetting the caudal angulation in the previous literatures [12,13,14,15].

The aims of this study are to clarify the gender differences of optimal S2AIS trajectories between males and females using three-dimensional computed tomography (3D-CT) analysis in the Japanese population, and to elucidate the true lateral angulation of S2AIS in the axial plane by resetting the caudal angulation of CT-imaging plane manipulation.

## 2. Materials and Methods

After the approval of our institution review boards, pelvic 3D-CTs of 50 consecutive subjects (25 male/25 female) who visited for abdominal aortic aneurysm or peripheral vascular disease were evaluated as a normal Japanese population. Subjects who had a history of any trauma and deformity in the sacropelvic lesion were excluded from this study. The scanned CT images were downloaded as DICOM images and converted into 3D-CT images using the software Ziostation2 (Ziosoft Inc., Tokyo, Japan). The starting points of S2AIS were determined on the 3D-CT images—2 mm inferior and 2 mm lateral to the S1 dorsal foramen bilaterally (right S2AIS, left S2AIS)—and bilateral posterior superior iliac spine (PSIS) were also determined on the 3D-CT images. The optimal trajectories of S2AISs were drawn using CT-imaging plane manipulation (caudal angulation) (Figure 1a–c), and the maximum length of bilateral S2AIS was measured. Bilateral S2AIS pathways were extracted on the 3D-CT images, and the minimal area of the bilateral S2AIS pathways was measured on the perpendicular plane of the S2AIS pathways (Figure 2a–c). After resetting the caudal angulation in CT-imaging plane manipulation, the trajectories of bilateral S2AISs in the axial plane (Figure 3a), sagittal plane (Figure 3b), and coronal plane (Figure 3c) were measured, and distances from the midline to the bilateral S2AISs starting points, distances from the midline to bilateral PSIS, and depths of the bilateral S2AISs starting points from skin were measured. All parameters were measured by one of the authors (H.F.), who is an experienced spine surgeon using S2AIS frequently, in this study.

The measurement values were summarized in this study (Table 1), and the average measurement values of the right and left side in both males and females were calculated, in order to compare with those in the previous literature (Table 2). Because previous studies evaluated the lateral angulation only in the transverse plane without resetting the caudal angulation, the lateral angulations of S2AISs in the axial plane were estimated using conversion Formula (1) (Figure 4):tanθ° = BC/AC = sinx°/cosx°cosy° = tanx°/cosy°(1)

The study complied with the declaration of Helsinki and was approved by the Ethics Review Committees (approval number: 20140392 in Keio University, and 20-Nr-013 in International University of Health and Welfare Narita Hospital).

### Statistical Analysis

All data were expressed as the mean ± standard deviation. The measurement values of the right and left side in both males and females for each parameter were compared with a two-way analysis of variance. Correlations between patients’ height and maximal length of S2AISs, and correlations between patients’ height and minimal areas of S2AIS pathways were determined using Pearson correlation coefficient. SPSS, version 21.0 (IBM Corp., Armonk, NY, USA), was used, and a *p*-value < 0.05 was considered statistically significant. Power analysis was performed using G*Power 3.1 (Heinrich Heine University, Düsseldorf, Germany). An effect size was 1.06, a type I error probability was 5%, and a type II error probability was 4.1% (i.e., power of 95.9%).

## 3. Results

Mean age (male/female) was 72.0 ± 7.7 (55–80)/72.8 ± 9.0 (57–88) years old, and patients’ height was 166.2 ± 8.1/151.2 ± 5.2 cm. Although there was no significant difference in age between males and females, there was a significant difference in height between males and females (*p* < 0.01). The results of measurement values were shown in Table 1. The mean maximal lengths of S2AIS were (right S2AIS/left S2AIS): 102.5 ± 6.5/102.3 ± 6.9 mm in males and 97.8 ± 4.9/95.9 ± 4.9 mm in females. The mean minimal areas of S2AIS pathways were (right S2AIS/left S2AIS): 5.6 ± 1.1/5.6 ± 1.1 cm^2^ in males and 4.8 ± 0.7/4.9 ± 0.7 cm^2^ in females. There were significant differences in the maximal lengths and minimal areas of S2AIS pathways between males and females (*p* < 0.01). There were positive correlations between patients’ height and the maximal length of S2AISs (right S2AIS; r = 0.64, *p* < 0.01/left S2AIS; r = 0.70, *p* < 0.01), and patients’ height and the minimal areas of S2AIS pathways (right S2AIS; r = 0.63, *p* < 0.01/left S2AIS; r = 0.65, *p* < 0.01). The mean insertion angles of S2AIS in the axial plane were (right S2AIS/left S2AIS): 45.3 ± 3.6/44.3 ± 3.2° in males and 47.7 ± 4.0/46.1 ± 3.6° in females. The mean insertion angles of S2AIS in the sagittal plane were (right S2AIS/left S2AIS): 37.5 ± 5.3/36.7 ± 4.0° in males and 36.5 ± 4.4/36.5 ± 4.5° in females. In addition, the mean insertion angles of S2AIS in the coronal plane were (right S2AIS/left S2AIS): 36.5 ± 3.0/37.0 ± 3.1° in males and 33.7 ± 4.6/34.5 ± 4.2° in females. There were significant differences in the axial and coronal trajectories of bilateral S2AISs between males and females (*p* < 0.05). There was no significant difference in the sagittal trajectories of bilateral S2AISs between males and females. Mean distances from the midline to starting points (right S2AIS/left S2AIS) were 27.9 ± 2.6/27.8 ± 2.3 mm in males and 26.0 ± 2.1/26.6 ± 1.9 mm in females. Mean distances from the midline to PSIS were 38.9 ± 4.1/38.8 ± 3.7 mm in males and 41.1 ± 4.2/41.3 ± 3.5 mm in females. Interestingly, the starting points of S2AIS tended to be shorter from the midline in females, whereas the PSIS was significantly longer from the midline in females (*p* < 0.05). The mean depths of S2AIS starting points from skin were (right S2AIS/left S2AIS): 38.7 ± 7.6/39.0 ± 7.1 mm in males and 39.7 ± 7.3/39.4 ± 7.3 mm in females. Although mean depths of S2AIS starting point tended to be deeper in females, they did not reach significance.

The average measurement values of the right and left side in both males and females are shown in Table 2. The estimated lateral angulations of S2AISs in the axial plane using our conversion formula became larger compared to the lateral angulation in the transverse plane depending on the amount of mean caudal angulation.

## 4. Discussion

Over the years, secure rigid sacropelvic fixation has posed challenges to spine surgeons. Although various sacropelvic fixation techniques have been applied [2,3,4,5,6,7,8,9,10], the S2AIS technique is the only available technique for percutaneous sacropelvic fixation to date that can provide an in-line connection to the proximal spinal constructs without using offset connectors [8,9]. Recently, the percutaneous pedicle screw technique has been widely applied to various spinal disorders with small skin incisions to avoid soft tissue dissections [16]. Several advantages of minimally invasive spine stabilization (MISt) with the percutaneous pedicle screw technique have been reported, such as less blood loss, lower infection rates, and cost effectiveness [17,18,19,20]. However, a minimally invasive sacropelvic fixation is not well established due to the difficulties of placing percutaneous anchors into the ilium and attaching them to the proximal screws. Because the percutaneous S2AIS technique is one of the most challenging percutaneous screw techniques, it is important for surgeons to evaluate optimal trajectories of S2AIS. S2AIS is usually placed with a freehand technique or image guidance using either intraoperative fluoroscopy or CT-based navigation. The accuracy of S2AIS placements using intraoperative CT-based navigation has been reported in conventional open spinal surgeries. Ray et al. [21] reported 18 patients who underwent posterior spinal fusion using intraoperative CT-based navigation, and 1 patient required repositioning of the screw because of an apparent breach in the anterior cortex in the ilium. Nottmeier et al. [22] reported 20 patients who underwent S2AIS placements using intraoperative CT-based navigation, and 5 screws penetrated the anterior cortex of the sacrum. Therefore, a misplacement of S2AIS can occur even in open conventional spinal surgery under CT-based navigation.

Although the S2AIS trajectories were evaluated in 3D-CT analysis in the previous literature, there have been only a few reports evaluating the trajectories of S2AIS according to gender [12,13,14,15]. Since the size and shape of the pelvis are different between males and females [11], optimal S2AIS trajectories should be evaluated regarding gender differences. Yamada et al. [13] reported that the lateral angulation of S2AIS in males (right 37.7°, left 37.9°) was significantly larger than in females (right 32.4°, left 32.8°), and the caudal angulation was significantly larger in females (right 33.9°, left 33.4°) than in males (right: 28.0°, left 27.5°) in Japanese patients with spinal disease. Zhu et al. [14] reported that the S2AIS trajectories in females (right 35.7°, left 34.5°) were more caudal than in males (right 30.0°, left 29.2°) in the sagittal plane, but the lateral angulation in the transverse plane (right 37.2°, left 36.5° in males; right 36.3°, left 35.7° in females) showed no difference between genders in a normal Chinese population. On the other hand, Wu et al. [15] reported that the lateral angulation of S2AIS increased (right 42.7°, left 41.9°), and the caudal angle decreased (right 27.4°, left 27.7°) in female adult degenerative scoliosis patients.

Optimal S2AIS trajectories were determined and drawn in the transverse plane using CT-imaging plane manipulation adjusting caudal angulation (Figure 1c); however, the lateral angulation of S2AIS should be measured in the axial plane after resetting the caudal angulation (Figure 3c). The lateral angulation becomes smaller in the transverse plane as the caudal angulation becomes larger (Formula (1), Figure 4). For example, the lateral angulation of 40° of S2AIS in the transverse plane would be approximately 44° in the axial plane, when the screw was angulated 30° caudally. Unfortunately, previous studies evaluated the lateral angulation only in the transverse plane without resetting the caudal angulation, and they have not shown the amount of caudal angulation in each their CT-imaging plane manipulation [12,13,14,15]; thus, it is impossible to calculate the exact lateral angulation of S2AIS in the axial plane. However, based on their results of mean caudal angulation of S2AIS, the mean lateral angulations of S2AISs in the axial plane were estimated using our conversion formula as shown in Table 2. Estimated lateral angulations of S2AISs in the axial plane became larger compared to the lateral angulation in the transverse plane depending on the amount of the caudal angulation. The lateral angulation of S2AIS in the transverse plane should be reset to the axial plane, because most surgeons stand up straight, and estimate the lateral angulation of S2AIS in the axial plane intraoperatively. Surgeons may mis-insert S2AIS more ventrally based on the measurement value in the transverse plane.

In this paper, we assessed S2AIS trajectories between males and females in a normal Japanese population. After resetting the caudal angulation in CT-imaging plane manipulation, the angulation of S2AIS in the axial plane was more lateral compared to previous reports [7,12,13,14,15]. In addition, the angulation of S2AIS was more lateral in the axial plane in females (right 47.7, left 46.1°) than in males (right 45.3, left 44.3°) and more horizontal in the coronal plane in females (right 33.7°, left 34.5°) than in males (right 36.5°, left 37.0°). Interestingly, our results show that mean distances from the midline to starting points of S2AIS tended to be shorter in females, whereas mean distances from the midline to the PSIS were significantly longer in females (Figure 5a,b). The differences of the anatomical shape of the pelvis between males and females might impact on the differences of axial and coronal S2AIS trajectories. It is suggested that women have a small sacrum, but a larger pelvic cavity. In fact, Kaufmann et al. [11] reported that the mean pelvic volume was 997 cm^3^ in men and 1093 cm^3^ in women. Furthermore, Baragi et al. [23] assessed a pelvic floor area between African and European American women, and they found that African American women had a 5.1% smaller pelvic floor area than European American women. The optimal S2AIS trajectories might be influenced by the differences of sex, race, ethnicity, and various spinal and hip diseases. Additionally, another possibility was the differences of insertion point of S2AIS that were determined by examiners. Ideally, screw trajectories should be determined by well-unified methods, and evaluated both in males and females in various situations.

In the present study, we also found that the maximal lengths and minimal areas of S2AIS pathways were significantly longer and larger in males, and there were positive correlations between patients’ height and the maximal lengths of S2AISs, and patients’ height and the minimal areas of S2AIS pathways. Therefore, a patient’s height can be useful as a reference in the determination of screw sizing as well as the anatomical shape of the pelvis.

This study has several limitations. First, this study targeted only the normal Japanese population. The results are limited by the possibility of selection bias. There might be differences in optimal S2AIS trajectories among various races, ethnicities, and spine and hip diseases. Another limitation is the measurement bias, which might have occurred due to the differences of insertion point of S2AIS that were determined by the examiners. An international, multi-center study with well-unified measurement methods is required in the future analysis.

## 5. Conclusions

In conclusion, this is the first comparative study of optimal S2AIS trajectories between males and females using 3D-CT analysis in a normal Japanese population considering the adjustment and reset of caudal angulation in CT-imaging plane manipulation. The angulation of S2AIS was more lateral in the axial plane in females and more horizontal in the coronal plane in females. There were positive correlations between patients’ height and the maximal lengths and minimal areas of S2AIS pathways; therefore, a patient’s height can be useful as a reference in the determination of screw sizing as well as the anatomical shape of the pelvis. Our results are useful not only for conventional open spinal surgery, but also for minimally invasive spine surgery.

## Figures and Tables

**Figure 1 jcm-11-02511-f001:**
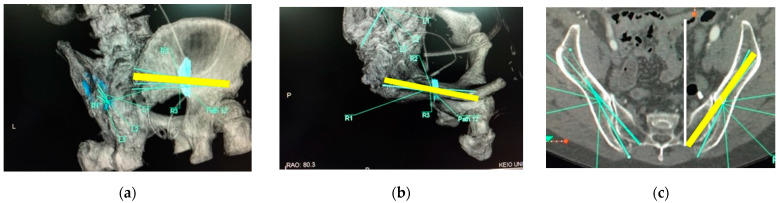
Optimal trajectories of S2AISs on the 3D-CT images. 3D-CT images were obtained, and the starting points of S2AIS were determined on the 3D-CT images; 2 mm inferior and 2 mm lateral to the S1 dorsal foramen bilaterally ((**a**,**b**) yellow lines). The optimal trajectories of S2AISs were drawn in the transverse plane using CT-imaging plane manipulation to adjust caudal angulation ((**c**) yellow line).

**Figure 2 jcm-11-02511-f002:**
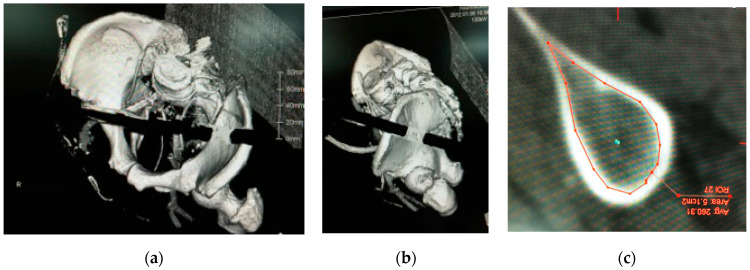
Minimal area of S2AIS pathway. Bilateral S2AIS pathways were extracted on the 3D-CT images (**a**,**b**), and the minimal area of bilateral S2AIS pathways were measured on the perpendicular plane of S2AIS pathways (**c**).

**Figure 3 jcm-11-02511-f003:**
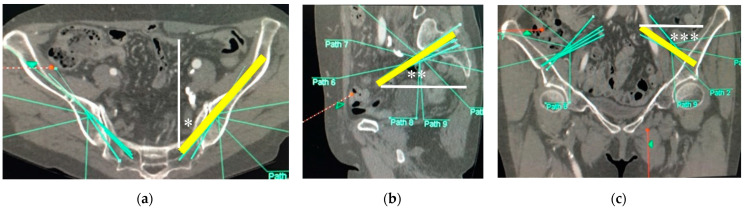
Trajectories of S2AISs. After resetting the caudal angulation in CT-imaging plane manipulation, the trajectories of S2AISs were shown as yellow lines (**a**–**c**), and the angles in the axial plane ((**a**), *), sagittal plane ((**b**), **), and coronal plane ((**c**), ***) were measured bilaterally.

**Figure 4 jcm-11-02511-f004:**
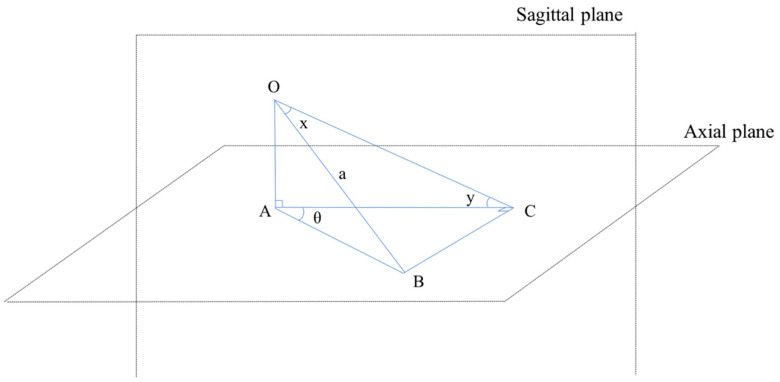
Conversion formula of the lateral angulation of S2AIS into the axial plane. The lateral angulation of S2AIS in the axial plane is calculated from that in the transverse plane using the following conversion formula: tanθ° = BC/AC = sinx°/cosx°cosy° = tanx°/cosy°. For example, the lateral angle of 40°of S2AIS in the transverse plane would be approximately 44° in the axial plane when the screw was angulated at 30° caudally. Points of A, B, and C are on axial plane. Points of O, A, and C are on the sagittal plane. ∠BOC = x°, ∠OCA = y°, OB = a, OC = a cosx°, AC = a cosx°cosy°, BC = a sinx°.

**Figure 5 jcm-11-02511-f005:**
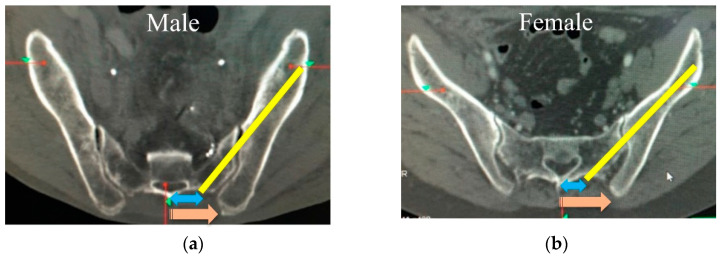
Schematics of the trajectories of S2AISs on males and females. The trajectories of S2AISs were shown as yellow lines (**a**,**b**). The starting points of S2AIS tended to be shorter from the midline in females, whereas the PSISs were significantly longer from the midline in females. The differences of the anatomical shape of the pelvis between males and females might impact on the differences of axial and coronal S2AIS trajectories (**a**,**b**).

**Table 1 jcm-11-02511-t001:** Measurement data of optimal S2AIS pathway in males and females.

	Male (*n* = 25)				Female (*n* = 25)					
	Right	SD	Left	SD	Right	SD	Left	SD	*p*-Value *	*p*-Value **
S2AIS pathway										
Maximum length (mm)	102.5	6.5	102.3	6.9	97.8	4.9	95.9	4.9	*p* < 0.01	*p* < 0.01
Minimum area (cm^2^)	5.6	1.1	5.6	1.1	4.8	0.7	4.9	0.7	*p* < 0.01	*p* < 0.01
S2AIS insertion angle										
Axial (°)	45.3	3.6	44.3	3.2	47.7	4.0	46.1	3.6	*p* < 0.05	*p* < 0.05
Sagittal (°)	37.5	5.3	36.7	4.0	36.5	4.4	36.5	4.5	*p* = 0.43	*p* = 0.87
Coronal (°)	36.5	3.0	37.0	3.1	33.7	4.6	34.5	4.2	*p* < 0.05	*p* < 0.05
Distance to the S2AIS insertion point from midline (mm)	27.9	2.6	27.8	2.3	26.0	2.1	26.6	1.9	*p* = 0.05	*p* = 0.15
Distance to the PSIS from midline (mm)	38.9	4.1	38.8	3.7	41.1	4.2	41.3	3.5	*p* < 0.05	*p* < 0.01
Distance to the S2AIS insertion point from skin (mm)	38.7	7.6	39.0	7.1	39.7	7.3	39.4	7.3	*p* = 0.62	*p* = 0.80

S2AIS, S2-alar-iliac screw; PSIS, posterior superior iliac spine; * *p*-value between males and females on the right; ** *p*-value between males and females on the left.

**Table 2 jcm-11-02511-t002:** Summary of the S2AIS pathways in the previous literature.

	Present Study		Chang et al.		Zhu et al.		Yamada et al.		Wu et al.	
Population	Japanese		American		Chinese		Japanese		Chinese	
Diagnosis	Normal		Normal		Normal		Spinal disease		Degenerative scoliosis	
Insertion point of S2AIS	2 mm inferior and 2 mm lateral to the S1 dorsal foramen		Not documented		1 mm inferior and 1 mm lateral to the S1 dorsal foramen		2 mm medial to apex of lateral sacral crest on midline between S1 and S2 dorsal foramen		The base of lateral sacral crest on the midline between S1 and S2 dorsal foramen	
Sex, Number	Male (*n* = 25)	Female (*n* = 25)	Male (*n* = 13)	Female (*n* = 7)	Male (*n* = 30)	Female (*n* = 30)	Male (*n* = 40)	Female (*n* = 40)	Male (*n* = 15)	Female (*n* = 25)
Maximum length (mm)	102.4	96.8	105.9	106.9	121.3	114.8	121.5	113.8	124.0	117.5
Caudal angulation = Sagittal angle (°)	37.1	36.5	36.7	41.6	29.2	34.5	27.5	33.4	28.0	27.6
Lateral angulation in the axial plane (°)	44.8	46.9	−	−	−	−	−	−	−	−
Lateral angulation in the transverse plane (°)	−	−	39.4	38.0	36.5	35.7	37.9	32.8	39.5	42.3
Estimated lateral angulation in the axial plane using conversion formula (°)	−	−	45.7	46.3	40.3	41.1	41.3	37.7	43.0	45.8

## Data Availability

Not applicable.

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
