# Peer review of "Comparative Study of S2-Alar-Iliac Screw Trajectories between Males and Females Using Three-Dimensional Computed Tomography Analysis: The True Lateral Angulation of the S2-Alar-Iliac Screw in the Axial Plane"

_jcm, 2022, doi:10.3390/jcm11092511_

Round 1

Reviewer 1 Report

Thank you to the editor for inviting me to participate in the review process for this article. This article is an imaging measurement report of the S2AI screw trajectory, which is not a very novel study as a whole. Considering that there are some ethnic differences in human anatomical parameters between different countries in the world, this article still has certain reference significance for Japanese orthopedic surgeons. However, some revisions need to be made before the article can be accepted.

In Table 2 and line 188, the author summarizes some parameters of the S2AI screw trajectory and their influencing factors in the previous literature. Clearly, these authors' studies did not all use the same entry point. I think the insertion point of the S2AI screw has a great influence on its optimal insertion angle. Therefore, I suggest that the author add the parameter of the screw entry point to Table 2. Second, the authors should conduct a comparative analysis in the discussion on how the relevant parameters would be affected if the S2AI screw entry point was changed for the same patient data.

The author mentions " In addition, the anglulation of S2AIS was more lateral in the axial plane in females (Right 47.7, Left 46.1°) than males (Right 45.3, Left 44.3°) and more horizontal in the coronal plane in females (33.7°, Left 34.5°) than males (36.5°, Left 37.0°). Interestingly, our results showed that mean distances from the midline to starting points of S2AIS tended to be shorter in females, whereas mean distances from the midline to the PSIS was significantly longer in females. These characteristics might impact the differences of axial and coronal S2AIS trajectories " on line 222-228. In my opinion this is a good entry point, but it requires the author to do a correlation analysis. It is suggested that the authors analyze the relationship between the distance between S2AIS insertion point from midline (or distance between PSIS insertion point from midline ) and the lateral angulation of the S2AI screw on the axial plane based on the existing data.

On lines 259-260, the authors conclude "therefore, patients' height would be a definitive factor for screw sizing", which I think is inappropriate. S2AI screw size should be determined by patient pelvic anatomical parameters rather than height.So I suggest to delete this statement "therefore, patients' height would be a definitive factor for screw sizing"

In the first paragraph of the Methods the authors mention "After the approval of our institution review boards, pelvic 3D-CTs of 50 consecutive subjects (25 male/25 female) who visited for abdominal aortic aneurysm or peripheral vascular disease were evaluated as a normal Japanese population".Considering that the journal of clinical medicine is a relatively authoritative journal, the author should clearly give the ethics approval number of the institution.

Author Response

Answer to Reviewers' comments

This manuscript was reviewed by the scientific reviewers of Journal of Clinical Medicine and many important comments have been provided. Because their comments were highly suggestive, we revised our manuscript according to their suggestions. Therefore, we would like to resubmit the revised manuscript and ask again for their review.

Reviewer #1

Comment 1: In Table 2 and line 188, the author summarizes some parameters of the S2AI screw trajectory and their influencing factors in the previous literature. Clearly, these authors' studies did not all use the same entry point. I think the insertion point of the S2AI screw has a great influence on its optimal insertion angle. Therefore, I suggest that the author add the parameter of the screw entry point to Table 2. Second, the authors should conduct a comparative analysis in the discussion on how the relevant parameters would be affected if the S2AI screw entry point was changed for the same patient data.

Answer 1: We appreciate the reviewer’s comments. We totally agree with the reviewer’s comment. We have added the parameter of the insertion point of the S2AI screw to Table 2. As the reviewer suggested, insertion point of the S2AI screw might have an influence on its optimal insertion angle. The optimal S2AIS trajectories might be influenced by the differences of sex, race, and various spine and hip diseases as well as the differences of insertion point of the S2AI screws as the reviewer suggested. We have added the sentences below in the discussion section and limitation.

Present study

Chang et al.

Zhu et al.

Yamada et al.

Wu et al.

Population

Japanese

American

Chinese

Japanese

Chinese

Diagnosis

Normal

Normal

Normal

Spinal disease

Degenerative

scoliosis

Insertion point of S2AIS

2 mm inferior and

2 mm lateral to the S-1 dorsal foramen

Not documented

1 mm inferior and

1 mm lateral to the S1 dorsal foramen

2 mm medial to apex of lateral sacral crest

on midline between S1 and S2 dorsal

foramen

The base of lateral sacral crest on the midline between S1 and S2 dorsal foramen

Sex, Number

Male (n=25)

Female (n=25)

Male (n=13)

Female (n=7)

Male (n=30)

Female (n=30)

Male (n=40)

Female (n=40)

Male (n=15)

Female (n=25)

Maximum length (mm)

102.4

96.8

105.9

106.9

121.3

114.8

121.5

113.8

124.0

117.5

Caudal angulation
= Sagittal angle (°)

37.1

36.5

36.7

41.6

29.2

34.5

27.5

33.4

28.0

27.6

Lateral angulation in the axial plane (°)

44.8

46.9

Lateral angulation in the transverse plane (°)

39.4

38.0

36.5

35.7

37.9

32.8

39.5

42.3

Estimated lateral angulation in the axial plane 

using conversion formula (°)

45.7

46.3

40.3

41.1

41.3

37.7

43.0

45.8

Table 2. Summary of S2AIS pathways in previous literatures.

The optimal S2AIS trajectories might be influenced by the differences of sex, race, ethnicity, and various spinal and hip diseases. And, another possibility was the differences of insertion point of S2AIS that were determined by examiners. Ideally, screw trajectories should be determined by well-unified methods, and evaluated both in males and females in various situations. (Line 250-255)

This study has several limitations. First, this study targeted only the normal Japanese population. The results are limited by the possibility of selection bias. There might be the differences of the optimal S2AIS trajectories among various race, ethnicity, and spine and hip diseases. Another limitation is the measurement bias, that might be occurred due to the differences of insertion point of S2AIS that were determined by the examiners. An international, multi-center study with well-unified measurement methods are required in the future analysis. (Line 260-266)

Comment 2: The author mentions " In addition, the angulation of S2AIS was more lateral in the axial plane in females (Right 47.7, Left 46.1°) than males (Right 45.3, Left 44.3°) and more horizontal in the coronal plane in females (33.7°, Left 34.5°) than males (36.5°, Left 37.0°). Interestingly, our results showed that mean distances from the midline to starting points of S2AIS tended to be shorter in females, whereas mean distances from the midline to the PSIS was significantly longer in females. These characteristics might impact the differences of axial and coronal S2AIS trajectories " on line 222-228. In my opinion this is a good entry point, but it requires the author to do a correlation analysis. It is suggested that the authors analyze the relationship between the distance between S2AIS insertion point from midline (or distance between PSIS insertion point from midline) and the lateral angulation of the S2AI screw on the axial plane based on the existing data.

Answer 2: We appreciate the reviewer’s comments and suggestion. Although we have analyzed the correlations between the distances of S2AIS and PSIS insertion points from midline and the lateral angulations of the S2AI screws on the axial plane, we did not find out significant correlations unfortunately. Because the pelvic cavity seems to be spread anteriorly in females (please see the Figure 5), the lateral angulation of S2AI screw might be influenced not only by insertion point of S2AI screw but also shape of the pelvis. Because our previous sentence might be confused, we have revised the sentences below in the Discussion section and Figure 5.

The differences of the anatomical shape of the pelvis between males and females might impact on the differences of axial and coronal S2AIS trajectories. (Line 244-246)

The differences of the anatomical shape of the pelvis between males and females might impact on the differences of axial and coronal S2AIS trajectories (a,b). (Line 270-271)

Comment 3: On lines 259-260, the authors conclude "therefore, patients' height would be a definitive factor for screw sizing", which I think is inappropriate. S2AI screw size should be determined by patient pelvic anatomical parameters rather than height. So I suggest to delete this statement "therefore, patients' height would be a definitive factor for screw sizing".

Answer 3: We appreciate the reviewer’s comments and suggestion. We totally agree with the reviewer’s comment. We have deleted the sentence "therefore, patients' height would be a definitive factor for screw sizing", and revised the sentence as below.

Therefore, a patient’s height can be useful as one of the references in determining screw sizing as well as the anatomical shape of the pelvis. (Line 259-260)

There were positive correlations between patients’ height and maximal lengths and minimal areas of S2AIS pathways, therefore, a patient’s height can be useful as one of the references in determining screw sizing as well as the anatomical shape of the pelvis. (Line 279-282)

Comment 4: In the first paragraph of the Methods the authors mention "After the approval of our institution review boards, pelvic 3D-CTs of 50 consecutive subjects (25 male/25 female) who visited for abdominal aortic aneurysm or peripheral vascular disease were evaluated as a normal Japanese population". Considering that the journal of clinical medicine is a relatively authoritative journal, the author should clearly give the ethics approval number of the institution.

Answer 4: We appreciate the reviewer’s suggestion. We have added the ethics approval numbers of our institutions in the Materials and Methods section as below as well as Institutional Review Board Statement.

The study complied with the declaration of Helsinki and was approved by the Ethics Review Committees (approval number: 20140392 in Keio University, and 20-Nr-013 in International University of Health and Welfare Narita Hospital). (Line 125-128)

The study was conducted according to the guidelines of the Declaration of Helsinki, and approved by the Institutional Review Board of Keio University (protocol code 20140392, November 26, 2020), and the Institutional Review Board of International University of Health and Welfare Narita Hospital (protocol code 20-Nr-013, April 27, 2021). (Line 289-292)

Reviewer 2 Report

Comparative Study of S2-Alar-Iliac Screw Trajectories Between Males and Females Using Three-Dimensional Computed Tomography Analysis: True Lateral Angulation of S2-Alar-Iliac Screw in the Axial Plane

The present study by Haruki Funao et al. aims to evaluate the different screw trajectories for sacropelvic fixation in a Japanese population.

Even if the topic is interesting and the measurements were made in a correct way, some points should be clarified:

  • It not clear why the authors considered normal subjects as inclusion criteria since the major interest of these measures are related to spinal pathology. The authors should clarify this point.
  • The use of one-way analysis of variance (ANOVA) is mainly linked to the presence of a categorical independent variable (with two or more categories) and a normally distributed dependent variable. The distribution of continuous variables should be verified. Moreover, a possible interaction between the two categorical variables (male/female and right/left side) could be analysed with a factorial ANOVA.
  • Table 2 is not presented in the results paragraph and in the methods of the article. I suggest to clarify in the methods how the authors obtained the previously published data and to compare their results with the articles by Chang and Zhu (normal patients). Moreover, I don’t understand the reason of reporting Yamada and Wu studies since they were made in patients with spinal pathology.
  • The measurements were performed only by one of the authors even if skilled in spinal surgery. This point should be underlined in the limitations since this method is prone to bias.

Author Response

Answer to Reviewers' comments

This manuscript was reviewed by the scientific reviewers of Journal of Clinical Medicine and many important comments have been provided. Because their comments were highly suggestive, we revised our manuscript according to their suggestions. Therefore, we would like to resubmit the revised manuscript and ask again for their review.

Reviewer #2:

Comment 1: It not clear why the authors considered normal subjects as inclusion criteria since the major interest of these measures are related to spinal pathology. The authors should clarify this point.

Answer 1: We appreciate the reviewer’s comment. As the reviewer suggested, we should analyze the screw trajectories of S2AIS focusing on various spinal diseases. Because there have been no reports that evaluate the optimal trajectories of S2AIS in the normal Japanese population, we would like to demonstrate the standard values of Japanese population in this study. We have added the sentences below in the limitations and conclusions.

This study has several limitations. First, this study targeted only the normal Japanese population. The results are limited by the possibility of selection bias. There might be the differences of the optimal S2AIS trajectories among various race, ethnicity, and spine and hip diseases. Another limitation is the measurement bias, that might be occurred due to the differences of insertion point of S2AIS that were determined by the examiners. An international, multi-center study with well-unified measurement methods are required in the future analysis. (Line 261-267)

In conclusions, this is the first comparative study of optimal S2AIS trajectories between males and females using 3D-CT analysis in a normal Japanese population considering the adjustment and reset of caudal angulation in CT-imaging plane manipulation. (Line 275-277)

Comment 2: The use of one-way analysis of variance (ANOVA) is mainly linked to the presence of a categorical independent variable (with two or more categories) and a normally distributed dependent variable. The distribution of continuous variables should be verified. Moreover, a possible interaction between the two categorical variables (male/female and right/left side) could be analysed with a factorial ANOVA.

Answer 2: We appreciate the reviewer’s comment. We apologize for the confusion. The measurements were compared between right and left side both in males and females using two-way ANOVA as the reviewer’s comment. We have revised the sentence below.

The measurements values of the right and left side in both males and females for each parameter were compared with two-way analysis of variance. (Line 130-132)

Comment 3: Table 2 is not presented in the results paragraph and in the methods of the article. I suggest to clarify in the methods how the authors obtained the previously published data and to compare their results with the articles by Chang and Zhu (normal patients). Moreover, I don’t understand the reason of reporting Yamada and Wu studies since they were made in patients with spinal pathology.

Answer 3: We appreciate the reviewer’s comment and suggestion. We have presented Table 2 in the methods and results section. We have calculated the average measurement values of the right and left side in both males and females in the present study and previous studies. Ideally, the optimal trajectories of S2AIS should be evaluated separately in the normal population and spinal pathology. Unfortunately, there has only been a few reports demonstrating the trajectories of S2AIS according to gender, even when we include both of the normal population and patients with spinal disorders. Therefore, we would like to demonstrate and compare the various measurement values from previous literatures in Table 2. We would really appreciate it if the reviewer could understand our thought.

The measurement values were summarized in this study (Table 1), and the average measurement values of the right and left side in both males and females were calculated, in order to compare with those in the previous literatures (Table 2). Because, previous studies evaluated the lateral angulation only in the transverse plane without resetting the caudal angulation, mean lateral angulation of S2AISs in the axial plane were estimated using conversion Formula 1. (Line 110-115)

The average measurement values of the right and left side in both males and females were shown in Table 2. Estimated lateral angulations of S2AISs in the axial plane using our conversion formula became larger compared to the lateral angulation in the transverse plane depending on the amount of mean caudal angulation. (Line 168-171)

Comment 4: The measurements were performed only by one of the authors even if skilled in spinal surgery. This point should be underlined in the limitations since this method is prone to bias.

Answer 4: We appreciate the reviewer’s comment and suggestion. We totally agree with the reviewer’s comment. The inter and intra variability were not assessed in most of previous literatures in this research area, the measurement by an experienced spinal surgeon should be appropriate. That said, this could be a bias as the reviewer suggested, we have added the sentence below in the limitation.

This study has several limitations. First, this study targeted only the normal Japanese population. The results are limited by the possibility of selection bias. There might be the differences of the optimal S2AIS trajectories among various race, ethnicity, and spine and hip diseases. Another limitation is the measurement bias, that might be occurred due to the differences of insertion point of S2AIS that were determined by the examiners. An international, multi-center study with well-unified measurement methods are required in the future analysis. (Line 261-267)

Round 2
